# Host Starvation and Female Sex Influence Enterobacterial ClpB Production: A Possible Link to the Etiology of Eating Disorders

**DOI:** 10.3390/microorganisms8040530

**Published:** 2020-04-07

**Authors:** Jonathan Breton, Justine Jacquemot, Linda Yaker, Camille Leclerc, Nathalie Connil, Marc Feuilloley, Pierre Déchelotte, Sergueï O. Fetissov

**Affiliations:** 1Inserm UMR 1073, Nutrition, Gut and Brain Laboratory, 76183 Rouen, France; Justine.jacquemot@gmail.com (J.J.); linda.yaker@hotmail.fr (L.Y.); pierre.dechelotte@chu-rouen.fr (P.D.); 2Institute for Research and innovation in Biomedicine, University of Rouen Normandy, 76000 Rouen, France; Leclerc.camille@hotmail.fr (C.L.); nathalie.connil@univ-rouen.fr (N.C.); marc.feuilloley@univ-rouen.fr (M.F.); 3Rouen University Hospital, CHU Charles Nicolle, 76183 Rouen, France; 4Laboratory of Microbiology Signals and Microenvironment, LMSM EA4312, F-27000 Evreux, France; 5Inserm UMR1239, Neuronal and Neuroendocrine Differentiation and Communication Laboratory, 76130 Mont-Saint-Aignan, France

**Keywords:** microbiota–brain axis, appetite, food intake, feeding behavior, Enterobacteriaceae, autoantibodies, sex differences, anorexia nervosa, bulimia, animal models, activity-based anorexia

## Abstract

Altered signaling between gut bacteria and their host has recently been implicated in the pathophysiology of eating disorders, whereas the enterobacterial caseinolytic protease B (ClpB) may play a key role as an antigen mimetic of α-melanocyte-stimulating hormone, an anorexigenic neuropeptide. Here, we studied whether ClpB production by gut bacteria can be modified by chronic food restriction and female sex, two major risk factors for the development of eating disorders. We found that food restriction increased ClpB DNA in feces and ClpB protein in plasma in both male and female rats, whereas females displayed elevated basal ClpB protein levels in the lower gut and plasma as well as increased ClpB-reactive immunoglobulins (Ig)M and IgG. In contrast, direct application of estradiol in *E. coli* cultures decreased ClpB concentrations in bacteria, while testosterone had no effect. Thus, these data support a mechanistic link between host-dependent risk factors of eating disorders and the enterobacterial ClpB protein production.

## 1. Introduction

Complex interactions between the host genome and bacterial metagenome may contribute to the risk factors to develop anorexia nervosa (AN) and bulimia nervosa (BN), two main forms of eating disorders (EDs) in humans [1]. A genetic predisposition for autoimmunity and its significant association with ED strongly support an autoimmune component in the mechanism of both AN and BN [2,3,4]. In this light, altered signaling between gut bacteria and their host has recently been implicated in the pathophysiology of EDs, whereas the enterobacterial caseinolytic protease B (ClpB) protein may play a key role as an antigen mimetic of α-melanocyte-stimulating hormone (α-MSH), an anorexigenic neuropeptide [5,6]. The proposed pathophysiological model is based on the autoimmune response to ClpB considering an important role played by the melanocortin (MC) system in the regulation of appetite and energy metabolism, whereas α-MSH activates the MC type 4 receptor (MC4R), inducing satiety and a negative energy balance [7]. In fact, *Escherichia* (*E*)*. coli* ClpB is a 96 KDa chaperon protein displaying an α-MSH-like motif and, therefore, has a property of an α-MSH antigen mimetic, triggering the production of α-MSH-cross-reactive antibodies [5]. The clinical relevance of α-MSH-reactive immunoglobulin (Ig)M and IgG antibodies to EDs was supported by correlations of their plasma levels with psychopathological traits in both AN and BN patients [8]. The mechanism of action of α-MSH-reactive IgG may include activation of MC4R by the immune complexes with α-MSH, which deregulates feeding behavior and emotions [9].

Considering the postulated etiologic role of ClpB in the pathophysiology of EDs, it is necessary to analyze its regulation by host-dependent behavioral and genetic risk factors of EDs. Chronic food restriction and female sex are two major risk factors of developing both AN and BN with the female/male ratios of 9 to 1 [10]. Thus, in the present study, we analyzed whether chronic food restriction may differentially regulate ClpB production by gut bacteria in male and female rats and tested the *in vitro* effects of estradiol and testosterone on ClpB production by *E. coli*. A sex-dependent response to starvation in rats on ClpB- and α-MSH-reactive IgG and IgM production was also verified.

## 2. Materials and Methods

### 2.1. Animal Model of Food Restriction

Animal care and experimentation complied with both French and European Community regulations (1986 Directive 2010/63/EU) and were approved by the local ethical committee (N 8690, 08/07/2019). The rat model of food restriction was consistent with that reported in the scientific literature [11]. Briefly, 2 sets of both male (*n* = 12) and female (*n* = 12) Sprague-Dawley rats (Janvier, Le Genest St Isle, France) were acclimatized in individual cages at 22 ± 1 °C for 4 days. During this period and for all experiments, the 12-h light-dark cycle was inverted (dark phase: 9:30 AM–9:30 PM). Seven days prior to the restricted time access to food, male and female rats had free access to water and standard diet. For both sexes, food access was limited to 1.5 h per day until the end of the experiment day 14); drinking water was always available *ad libitum*. Food was given at the beginning of the dark phase. Food consumption was measured when food was removed. Body weight, food intake, and water intake were recorded daily during the protocol from day (D)-7 to D-14.

### 2.2. Fecal and Tissue Sampling

Rat feces were sampled at D-2 and D-14 of the protocol, directly frozen in liquid nitrogen, and stored at −30 °C prior to starting the DNA and Biotyper analyses. Similarly, plasma was collected twice (D-2 and D-14) by a puncture from the retroorbital sinus of rats, spun at 1480× *g* for 15 min at 4 °C, and then immediately frozen at −80 °C. At the end of the protocol (D-14), rats were euthanized, and different parts of the intestinal tract were dissected; the mucosal layer was scrubbed and frozen in liquid nitrogen, and then stored at −30 °C before ClpB assay.

### 2.3. Identification of Bacteria by MALDI-TOF MS Biotyper

Bacterial strains from the fecal microbiota of male and female rats, before and after restriction, were isolated on Luria-Bertani medium and identified by analysis of the total proteome using an Autoflex III Matrix-Assisted Laser Desorption/Ionization-Time-Of-Flight mass spectrometer (MALDI-TOF MS) (Bruker, Marcy-l’Etoile, France) coupled to the MALDI-Biotyper 3.1 system, as previously described [12,13]. Formic acid was used on the bacterial spots as a quick extraction procedure [14], then the MALDI target plate was introduced in the mass spectrometer for measurement and data acquisition. For each sample, 600 spectra were pooled, and the generated spectra were compared with the MALDI-Biotyper 3.1 database. A score was calculated based on the matching between the reference spectrum and the unknown spectrum. A score of ≥2.0 allows species identification [15].

### 2.4. ClpB DNA Analysis

Quantitative polymerase chain reaction (qPCR) was performed to analyze the bacterial density of ClpB DNA using a CFX 96 q-PCR instrument (BioRad, Hercules, CA, USA). Total DNA was extracted from the rat feces using a QIAamp Fast DNA stool kit (QIAGEN Valencia, CA, USA) and quantified with a Nanodrop 2000c spectrophotometer (Nanodrop technologies, Wilmington, DE, USA). The qPCR mix included 5 μL of SYBR Green Master (QIAgen, West Sussex, UK), 0.5 μM each of the forward and reverse primers, DNA from samples (7 ng/µL), and water to give a total volume of 10 μL. The primer sequences were: Forward, 5′-GCAGCTCGAAGGCAAAACTA-3′, and reverse 5′-ACCGCTTCGTTCTGACCAAT-3′. The primers were purchased from Invitrogen (Cergy-Pontoise, France). A three-step PCR was performed for 40 cycles. The samples were denatured at 95 °C for 10 min, annealed at 60 °C for 2 min, and extended at 95 °C for 15 s as previously described [16].

### 2.5. Protein Extraction

Total proteins from both the mucosal tissue of the different parts of the intestinal tract and from *E. coli* K12 bacteria were extracted in 1 mL of PBS containing 1% of protease inhibitor (Sigma, St. Louis, MO, USA) and homogenized by sonication for 30 s at 4 °C. To separate proteins from the undissolved cell fragments, the bacterial homogenate was centrifuged at 4 °C for 30 min at 20,000× *g*. The supernatants containing proteins were collected and protein concentrations were measured using a Pierce BSCA protein assay kit (Thermo Scientific, Waltham, MA, USA).

### 2.6. ClpB ELISA

The concentration of ClpB protein was measured in plasma, the intestinal mucosa, and *E. coli* cultures by the enzyme-linked immunosorbent assay (ELISA) as previously described in detail [16]. For this purpose, 3 antibodies were used: Rabbit polyclonal anti-*E. coli* ClpB antibodies (Delphi Genetics, Gosselies, Belgium) as a capture antibody, mouse monoclonal anti-*E. coli* ClpB antibodies as a detection antibody (Delphi Genetics), and an alkaline phosphatase-conjugated goat anti-mouse revelation antibody (Jackson ImmunoResearch, Cambridgeshire, UK). The optical density of the ELISA reaction was measured at 405 nm using a microplate reader Metertech 960 (Metertech Inc., Taipei, Taiwan) and the ClpB concentration was determined using a ClpB protein (Delphi Genetics) standard curve ranging from 0 to 5, 10, 25, 50, 70, 100, and 150 pM. The optical density (OD) was determined at 405 nm using a microplate reader (Metertech Inc.). Each determination was performed in duplicate.

### 2.7. ClpB- and α-MSH-Reactive Antibody Assay

Plasma levels of anti-ClpB and α-MSH-reactive IgG and IgM were measured using ELISA according to a published protocol [17]. Briefly, α-MSH peptide (Bachem AG, Bubendorf, Switzerland) or ClpB protein (Delphi Genetics) was coated onto 96-well Maxisorp plates (Nunc, Rochester, NY, USA) using 100 µL and a concentration of 2 µg/mL in 0.5 M NaCO_3_, and 0.5M NaHCO_3_ buffer, pH 9.6, for 48 h at 4 °C. The plates were washed (×3) in phosphate-buffered saline (PBS) with 0.05% Tween 200, pH 7.4, and then incubated for 3 h at 37 °C with 100 μL of plasma diluted 1:400 in PBS pH 7.4 or in PBS with 3M NaCl + 1.5M glycine pH 8.9 for the assay of free or total antibody levels, respectively. The plates were washed (3×) and incubated with 100 μL of alkaline phosphatase-conjugated anti-rat IgG and IgM antibodies in PBS (1:2000, Jackson ImmunoResearch Laboratories). Following washing (3×), 100 μL of p-nitrophenyl phosphate solution (Sigma) were added as a substrate. After 40 min of incubation at room temperature, the reaction was stopped by adding 3N NaOH. The OD was determined at 405 nm using a microplate reader (Metertech Inc.). Blank OD values resulting from the reading of plates without the addition of plasma samples were subtracted from the sample OD values. Each determination was done in duplicate. The variation between duplicate values was less than 5%.

### 2.8. Bacterial Culture

*E. coli* K12 bacteria were cultured at 37 °C in 40 mL of Mueller Hinton (MH) medium (Becton, Dickinson, MD, USA) containing 30% beef infusion, 1.75% casein hydrolysate, and 0.15% starch with pH 7.3 at 25 °C in 50-mL Falcon vials. For the modeling of two scheduled daily meals in humans, bacteria received new MH medium every 12 h during 5 consecutive days as explained in Breton et al. [16]. Briefly, at the end of each 12-h cycle, bacteria were centrifuged for 5 min at 6000 rpm at room temperature (RT). The supernatants were discarded and replaced by an equivalent volume (~40 mL) of a new MH medium. Bacteria received either an acute (together with the last supplementation of MH medium) administration of a range of 17-β estradiol (0, 2, 20, and 200 pg/mL, and 12.5 ng/mL) or testosterone (0, 0.08, 0.8, 8.0, and 125 ng/mL) or chronic administration (at each supplementation with MH medium) of 12.5 ng/mL of 17-β estradiol. After the last supplementation of MH medium, bacterial cultures were centrifuged at 4 °C for 30 min at 4000× *g* and both pellets (containing bacteria) and supernatants (containing bacterial secretions) were sampled for protein extraction in the exponential phase (10 min after the supplementation of MH medium).

### 2.9. Immunocytochemistry

*E. coli* K12 cultures and ∆ClpB mutant cultures were spun down at 3200× *g* during 5 min at 4 °C and pellets were dissolved in 1 mL of PBS. Then, 20 µL of the bacterial solution were spread on slides, fixed by ethanol, and dried at 37 °C for 15 min. Immunocytochemistry was performed using triton solution (1%) to permeabilize bacteria followed by blocking solution (PBS, 5% Bovine serum albumin, 1% Triton solution, 0.2% Sodium Azide). Mouse monoclonal antibody against ClpB (1:50, Delphi Genetics) was incubated overnight at 4 °C and revealed with a secondary anti-mouse antibody (1:200) conjugated to fluorescein isothiocyanate (FITC) (Jackson ImmunoResearch) for 1.5 h at room temperature. Two drops of 4′,6-diamidino-2-phenylindole (DAPI) (Sigma) were added to each slide to counterstain the bacterial DNA. Microscopy was performed and images were taken using a Zeiss Imager Z1 fluorescence microscope equipped with an Apotome and an AxioCam digital camera (Zeiss, Oberkochen, Germany). Immunopositive cells were counted at 100× magnification from 5 different slides per group.

### 2.10. Total RNA Extraction

*E. coli* K12 total RNA was extracted in cold TRIZOL reagent (Invitrogen, Carlsbad, CA, USA). After extraction, RNA concentrations were measured using a NanoDrop spectrophotometer. A reverse transcription reaction was performed to generate cDNA with 1 µg of total RNA using of M-MLV reverse transcriptase (200 U/µL) (ThermoFisher, Carlsbad, CA, USA). Quantitative polymerase chain reaction (PCR) was performed on all samples using a BioRad CFX96 Real Time PCR System and SYBR Green Master Mix (Life Technologies, Carlsbad, CA, USA). The relative levels of ClpB mRNA expression were estimated by the inverse values of the amplification cycle threshold (Ct) for each cDNA sample curves.

### 2.11. Data and Statistical Analysis

Results were analyzed using GraphPad Prism 5.02 (GraphPad Software Inc., San Diego, CA, USA). Normality was evaluated by the Kolmogorov–Smirnov test. Group differences were analyzed by the analysis of variance (ANOVA) or the non-parametric Kruskal–Wallis (K-W) test with the Tukey’s or Dunn’s post-tests, according to the normality results. Where appropriate, individual groups were compared using the Student’s t-test or the Mann-Whitney (M-W) test according to the normality results. Data are shown as means ± standard error of means (SEM), and for all tests, *p* < 0.05 was considered statistically significant.

## 3. Results

### 3.1. Rat Model of Food Restriction

Both male and female rats had restricted time access to food for 1.5 h per day for 14 days (Figure 1A). During food restriction, rats progressively lost body weight, reaching −25% in males and −24% in females at the last day of restriction (Figure 1B). Food intake during restriction was similar in male and female rats, who gradually adapted to the restricted access to food by doubling their 1.5-h food intake from the first to the last day of restriction (Figure 1C). A decrease of −40% and −50% of food intake was recorded at the last day of restriction as compared to the day before the restriction in female and male rats, respectively. Drinking water was always available *ad libitum* and water intake slightly declined in both male and female rats during restriction following the adaptation to the restricting feeding schedule (Figure 1D).

### 3.2. Effects of Food Restriction on ClpB Production

*E. coli* ClpB DNA was detected in the feces of all animals and its expression was increased during food restriction as compared to the basal condition in both male and female rats (Figure 2A). No significant difference in ClpB DNA expression levels was found between sexes neither in basal nor in food-restricted conditions. Using the “Biotyper” bacterial protein chromatography technology, we confirmed that *E. coli* bacteria were present in the majority of rats of both sexes before restriction and each rat displayed at least one colony of Enterobacteriaceae after food restriction, i.e., ensuring the enterobacterial ClpB production (Figure 2B). 

After food restriction, ClpB protein concentrations were analyzed in the mucosa of different parts of the rat intestine, including the duodenum, jejunum, ileum, and colon. ClpB protein was detectable in all parts of the intestine in both males and females with region- and sex-specific differences (Figure 2C). As such, ClpB concentrations in females were lower in the jejunum but increased in the lower intestine, including both the ileum and colon (Figure 2C).

ClpB protein was detected at picomolar concentrations in the plasma of all animals and it was increased after food restriction in both male and female rats (Figure 2D). Of notice, plasma ClpB was higher in female vs. male rats prior to food restriction, but it was not different between sexes after food restriction (Figure 2D). Plasma levels of ClpB correlated significantly with fecal ClpB DNA content only in males (Figure 2E).

We also measured the ClpB protein concentrations in plasma samples available from our previous study in male mice with activity-based anorexia (ABA), a rodent model of anorexia nervosa, based on the combination of restricted feeding with physical activity using a running wheel [18]. We found that plasma ClpB concentrations were increased in the ABA mice, i.e., the group that lost body weight due to starvation and physical activity, as compared to ad libitum-fed control mice (Figure 2F).

### 3.3. Effects of Food Restriction on ClpB- and α-MSH-Reactive Immunoglobulin Production

Plasma levels of ClpB- and α-MSH-reactive IgG and IgM were analyzed before and after food restriction using ELISA in normal and dissociative buffer conditions, corresponding to the free or total Ig assay, respectively [17]. Both free and total levels of ClpB- and α-MSH-reactive IgG were increased after food restriction in female rats (Figure 3A–D). In male rats only, total α-MSH-reactive IgG was increased while ClpB-reactive IgG was not significantly affected by food restriction (Figure 3A–D). 

While food restriction did not significantly affect ClpB-reactive IgM, the total level was higher in females vs. males in basal conditions (Figure 3E,F). A tendency of increased levels of free α-MSH-reactive IgM was found in both sexes and it was significant for total α-MSH-reactive IgM, with a higher response by female rats (Figure 3G,H). By analyzing the correlations between IgG and IgM levels and food intake during the last day of food restriction, a significant correlation was found for free ClpB-reactive IgG (Pearson’s *r* = 0.36, *p* < 0.05).

### 3.4. Effects of Sex Hormones on In Vitro ClpB Production

To determine a possible direct effect of male and female sex hormones on ClpB production, we acutely applied different physiological doses of testosterone (0.08–125 ng/mL) and 17-β estradiol (2.0 pg/mL−12.5 ng/mL) on *E. coli* cultures and measured ClpB concentrations in bacterial homogenates 10 min after the hormone’s application, i.e., during the bacterial exponential growth phase. We found that any tested doses of testosterone did not have a significant effect on ClpB *in vitro* production (Figure 4A). In contrast, estradiol decreased the ClpB protein concentration at its highest dose and its lower doses displayed similar tendencies (Figure 4B).

To get further insight into the effects of estradiol on ClpB production, we used immunocytochemistry to visualize ClpB in bacterial cells (Figure 4C). The specificity of the ClpB immunostaining was confirmed by the absence of the immunofluorescence in ClpB-deficient *E. coli* as a negative control, and its abundance in ClpB-overexpressing *E. coli* as a positive control (Figure 4C, Delta “-“, and Delta “+”, respectively). We found that the relative number of ClpB-positive vs. total bacterial cells was decreased after treatment with estradiol at both the lowest and highest doses, while the intermediate doses showed decreasing tendencies (Figure 4D). Moreover, the percentage of ClpB-positive cells correlated positively with the ClpB concentrations measured in the bacterial culture medium (Figure 4E).

Chronic application of estradiol on *E. coli* cultures resulted in decreased ClpB protein levels in both bacterial homogenates and in culture medium (Figure 4F,G), underlying a decreased rate of ClpB secretion (Figure 4H). We did not find a significant effect of estradiol on ClpB mRNA expression (Figure 4I).

## 4. Discussion

The present study revealed that enterobacterial ClpB production is influenced by host-dependent factors, including starvation and female sex, two major risk factors for the development of eating disorders. A possible role of the gut microbiota–immune–brain axis in the origin of ED as well as neuropsychiatric and autoimmune disorders has recently been discussed, emphasizing the presence of gut dysbiosis and its interactions with the immune and neuroendocrine systems [19,20,21]. Indeed, an altered composition of gut microbiota in AN patients has been reported in several studies and it was associated with psychological problems [22,23,24]. A recent study also reported that the colonization of germ-free mice with fecal microbiota from AN patients was accompanied by lower food intake and body weight gain than in mice receiving microbiota from healthy normal weight donors [25]. These data provide a proof of concept that gut bacteria in AN patients produce some molecules with enhanced satietogenic effects on the host. Such a possibility is in line with the postulated physiological role of gut microbiota in the regulation of host appetite [26]. For instance, we previously found that ClpB production is enriched in the protein fractions of *E. coli* from the stationary growth phase induced by regular nutrient provision, which activates intestinal and brain satiety pathways [16]. Moreover, ED patients display elevated plasma levels of enterobacterial ClpB [27], while an increased prevalence of Enterobacteriaceae was reported in AN patients [28,29,30]. Since ClpB-expressing *E. coli* display anorexigenic and body weight-lowering properties that are dependent on ClpB’s presence [5], it is possible that an increased prevalence of ClpB-producing bacteria may underlie the enhanced anorexigenic effect of gut bacteria in ED patients. Therefore, understanding the host-related factors regulating ClpB production may help to clarify the etiology and pathogenesis of ED.

Voluntary or imposed chronic and severe food restriction is probably the main risk factor for the development of ED. It should not be confounded with caloric restriction not leading to malnutrition and known to improve the health and lifespan in most organisms, including monkeys [31]. In contrast, severe food restriction is stressful for the host, leading chronically to malnutrition; it activates catabolic processes and increased hunger sensation. Such a host-related condition is imposed on the gut microbiota, which adapt to the host energy deficit by changing the composition and metabolic processes [18,32,33]. The present study showed that chronically food-restricted rats of both sexes increased ClpB DNA, reflecting an increase of ClpB-producing bacteria. Increased ClpB DNA was accompanied by increased plasma levels of ClpB, supporting an increase of its systemic and immunogenic effects. Increased plasma ClpB levels were also found in starved ABA mice. The reason for increased ClpB gene and protein expression during host starvation can be related to the need of an increased protein disaggregation function of this chaperon protein during various catabolic processes and enzymatic remodeling aimed at protecting the survival of the bacterial population [34]. Increased ClpB production itself is not pathogenic and can be, instead, health protective [35,36]. It will be of interest to test whether ClpB also may act as a caloric restriction mimetic, i.e., a substance igniting the protective effects of caloric restriction [37]. The direct satietogenic effect of ClpB as an α-MSH mimetic may help the host to reduce the hunger sensation during starvation. Nevertheless, rats adapt themselves to the restricted feeding by acute hyperphagia during the short period of time when food is available. This adaptation may become possible due to fast eating without satiety sensation. We may speculate that increased levels of anti-ClpB antibodies may at least partly neutralize increased ClpB production and, hence, reduce ClpB- and α-MSH-induced satiety signaling. In fact, food intake in the last day of restriction correlated positively with plasma levels of anti-ClpB IgG. Moreover, a previous study showed that an increase of anti-ClpB antibodies induced by ClpB immunization in mice was associated with increased food intake and meal size [5].

Our ClpB immunodetection test was based on the use of antibodies raised against *E. coli* ClpB protein, the structure of which is highly conserved in the order of Enterobacteriales [36]. It is hence likely that ClpB concentrations in the gut and plasma of our study reflect enterobacterial ClpB production. We cannot, however, exclude that it relates partly to other bacterial taxa. Using the “Biotyper” technology, we confirmed the presence of Enterobacteriaceae in all studied animals after food restriction, which most likely underlies the ClpB concentrations measured in the gut and plasma. 

It is of interest that ClpB was not equally present in different parts of the gut. In male rats, it was more abundant in the duodenum and colon than in the ileum, while in females, it was low in the jejunum but was more abundant in other parts, in particular in the colon. The functional significance of the ClpB regional distribution is presently unclear, but we can speculate that it may activate different types of the enteroendocrine cells present alongside the different parts of the intestine. As such, the presence of ClpB protein in the duodenal mucosa of both male and female rats raises the question of its possible interaction with cholecystokinin, a short-term satiety hormone produced in the duodenum [38]. A high level of ClpB detected in the colonic mucosa of female rats may be relevant to the secretion of peptide YY (PYY), a long-term satiety hormone synthetized in the lower gut [39]. In fact, ClpB was shown to dose-dependently stimulate PYY secretion by *in vitro* primary cultures of the rat colonic mucosa [40]. Increased PYY plasma levels were also reported in AN patients [41]. 

The study found sex differences of the ClpB plasma level, which was elevated in female rats at the basal condition. Food restriction increased ClpB levels in both males and females so that no more significant differences were observed between sexes after restriction. Considering the similar basal levels of ClpB DNA, we cannot attribute the increased plasma ClpB to the increased ClpB gene prevalence in females. In fact, although increased prevalence of *E. coli* may sometimes be detected in female rats [42], in the present study, 10 out 12 male rats and 11 out of 12 female rats had *E. coli* in their microbiota. Therefore, the increased plasma ClpB in females of this study must be related to other regulatory factors, e.g., to increased basal plasma levels of ClpB-reactive IgM. Being a secretory immunoglobulin, IgM may increase the transportation of ClpB across the gut barrier similar to some IgA in their bacterial translocation functions [43]. Moreover, we found that plasma ClpB levels correlated directly with fecal ClpB DNA in male but not female rats. We may speculate that this phenomenon is related to a more variable ClpB neutralizing immune response in females vs. males, e.g., females characterized by increased basal levels of total immunoglobulins [42], the production of which is dependent on the estrous cycle [44,45]. We cannot exclude either the existence of sex differences at the level of ClpB transport across the intestinal barrier. Thus, sex-related differences expressed as both elevated plasma ClpB protein and ClpB-reactive IgM at basal conditions in female rats may at least partly contribute to the understanding of the female sex as a risk factor for eating disorder.

In fact, after an increase of ClpB expression during starvation, the immune system in females is already sensitized for the ClpB antigen to raise an adaptive antibody response, as expressed in the increased production of anti-ClpB IgG. While such IgG are not pathogenic themselves, their cross-reactivity with α-MSH may lead to an increased production of immune complexes, with α-MSH overactivating the MC4R and resulting in increased satiety and anxiety of ED patients [9]. In the present study, an increased α-MSH-reactive IgM and IgG response to starvation was found in both sexes, but it was more pronounced in females. Thus, considering the key postulated role of the pathogenic α-MSH-reactive IgG in the pathophysiology of ED [6], starvation- and female sex-induced increased production of ClpB antigen and ClpB-reactive antibody, respectively, may constitute the molecular events contributing to these two risk factors of ED.

In this study, we also tested the possible direct effects of sex hormones on ClpB production by *E. coli in vitro* as a model of their possible action on gut bacteria *in vivo*. In fact, sex hormones are naturally present in the gut and play a physiological role in the regulation of microbiome action on the host [46]. No significant effect was observed for testosterone, although a tendency of an increase of ClpB was observed with its highest dose. In contrast, both acute and chronic application of estradiol decreased ClpB concentrations in bacteria. Since no changes in ClpB mRNA were detected, the observed changes in ClpB protein concentrations may be due to post-translational regulatory mechanisms. It is possible that the estradiol-induced decrease of ClpB immunoreactivity is related to the increased formation of molecular complexes of ClpB with other chaperon proteins, which prevents its detection by antibodies. Such a possibility would imply that estradiol may increase protein aggregation in bacteria. In the mammalian host, estrogen triggers complex cellular responses via its receptors alpha and beta [47]. The mechanisms of a possible direct action of estrogen on bacteria need further clarification; for instance, estradiol may compete with the quorum-sensing N-Acyl homoserine lactone binding to its receptors in Gram-negative bacteria, resulting in downregulation of the quorum-sensing-regulated genes [48]. A link between quorum sensing and ClpB expression is also known [49]. Whether increased estrogen secretion in the host may decrease ClpB plasma levels with functional implications for the host appetite remains to be studied. For instance, puberty in girls is typically accompanied by an increased appetite [50], which in some individuals may cause restrictive feeding behavior and, hence, may trigger risk factors to develop ED. Thus, a possible inhibition of ClpB production in puberty by estrogens may play an indirect role in the mechanisms of etiology of ED, which often start during adolescence [51].

## 5. Conclusions

In conclusion, the present study, using a model of chronic food restriction in rats, revealed that host starvation stimulates the production of ClpB in the gut as well as the levels of ClpB protein in the plasma in both males and females. Thus, while non-pathogenic itself, increased bacterial ClpB antigen may become a risk for an autoimmune response in genetically predisposed individuals. Furthermore, female sex, another major risk factor of ED, was associated with increased plasma ClpB and ClpB-reactive antibodies, while estradiol decreased ClpB concentrations *in vitro*. These data further support a mechanistic link between this bacterial protein and the autoimmune etiology of ED.

## Figures and Tables

**Figure 1 microorganisms-08-00530-f001:**
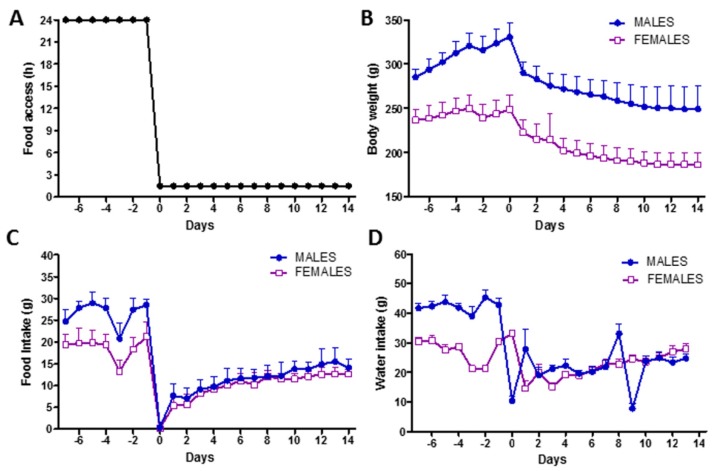
Rat model of chronic food restriction. (**A**) Experimental design showing the food access per day. (**B**) Body weight dynamics. (**C**) Daily food intake dynamics. (**D**) Daily water intake dynamics.

**Figure 2 microorganisms-08-00530-f002:**
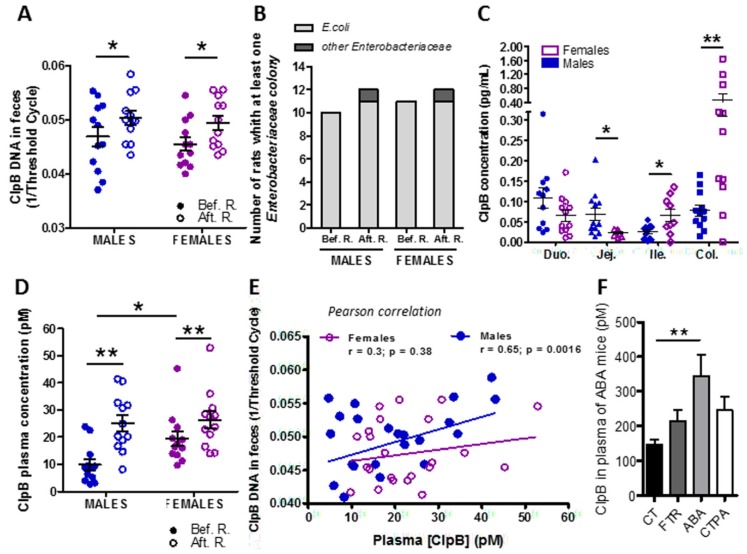
Effects of food restriction on ClpB production. (**A**) Fecal ClpB DNA levels before (solid circles) and 14 days after (empty circles) food restriction in male and female rats. (**B**) Biotyper identification of Enterobacteriaceae in fecal microbiota before (Bef.) and after (Aft.) restriction (R.) in female and male rats. (**C)** ClpB protein concentration in the intestinal mucosa at the levels of the duodenum (Duo), jejunum (Jej), ileum (Ile), and colon (Col) in male (solid symbols) and female (empty symbols) rats after food restriction. (**D**) ClpB protein plasma concentration before (solid circles) and 14 days after (empty circles) food restriction in male and female rats. (**E**) Correlations between fecal ClpB DNA and plasma ClpB after food restriction in male (solid circles) and female (empty circles) rats. (**F**) Plasma levels of ClpB protein in male mice with activity-based anorexia (ABA) as compared to ad libitum-fed controls (CT) and feeding time-restricted (FTR) mice, plasma samples available from a previous study [18]. (**A**) 2-way ANOVA, time interaction * *p* = 0.02. (**C**) Students t-test * *p* < 0.05, ** *p* < 0.01. (**D**) 2-way ANOVA, time (** *p* = 0.001) and row Factor (* *p* = 0.04). (**F**) 1-way ANOVA, Bonferroni’s post-test * *p* < 0.05, ** *p* < 0.01.

**Figure 3 microorganisms-08-00530-f003:**
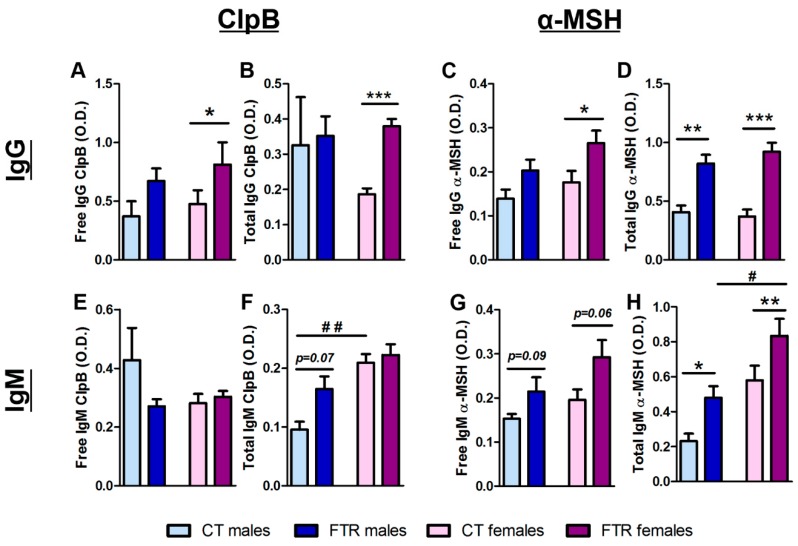
Effects of food restriction on ClpB- and α-MSH-reactive immunoglobulin production. ClpB- (**A**,**B**,**E**,**F**) and α-MSH-(**C**,**D**,**G**,**H**) reactive free and total IgG (**A**–**D**) and IgM (**E**–**H**) before (CT) and after 14 days of feeding time restriction (FTR) in male (light and dark blue, left side of each panel) and female (light and dark pink, right side of each panel. **A**–**D**,**F**,**H**, Students t-test * *p* < 0.05, ** *p* < 0.01, *** *p* < 0.001, 1-way ANOVA, Bonferroni’s post-test # *p* < 0.05, ## *p* < 0.01.

**Figure 4 microorganisms-08-00530-f004:**
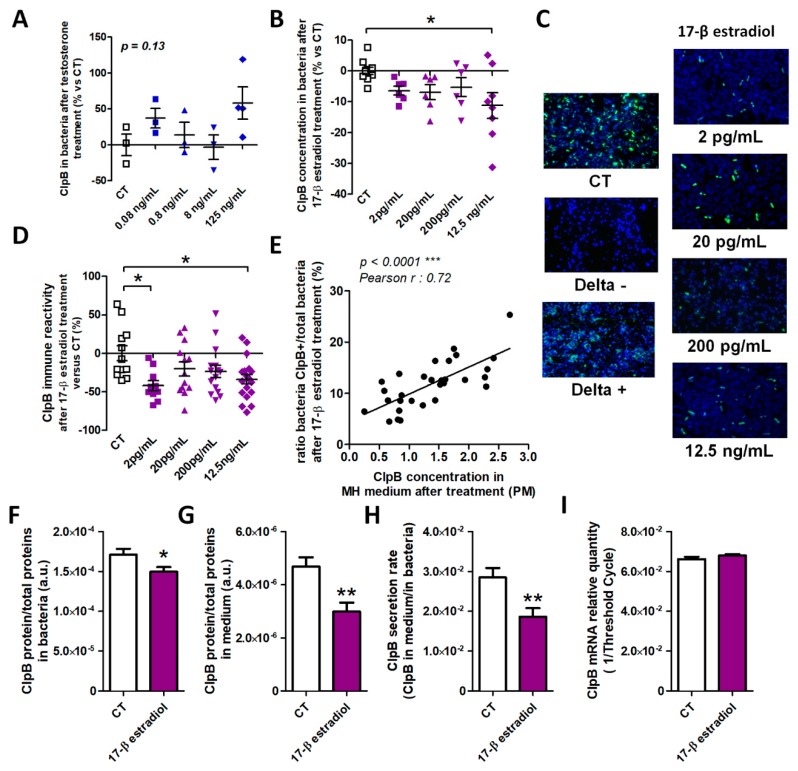
Effects of sex hormones on in vitro ClpB production. Changes of ClpB levels vs. control (CT) after acute application of several doses of testosterone (**A**) or estradiol (**B**) on *E. coli* cultures. (**C**) Immunocytochemical detection (green fluorescence) of ClpB in *E. coli* cultures treated with 4 doses of estradiol and in non-treated control (CT) bacteria as well as in ClpB-negative (Delta-) and -positive (Delta +) *E. coli* mutant strains, DAPI counterstaining. Magnification x 100 (**D**) Percentage of changes of ClpB-positive vs. total bacteria after estradiol treatment as compared to non-treated control (CT) bacteria and (**E**) their correlations with ClpB concentrations in culture medium. Effects of chronic application of estradiol on *E. coli* cultures on ClpB concentrations in bacteria (**F**) and culture medium (**G**) as well as on the ClpB secretion rate (**H**) and ClpB mRNA levels (**I**). **A**,**B**,**D**, 1-way ANOVA, Bonferroni’s post-test **p* < 0.05, ***p* < 0.01, **F**,**G**,**H**, Students t-test **p* < 0.05, ***p* < 0.01.

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
