# Peer review of "Host Starvation and Female Sex Influence Enterobacterial ClpB Production: A Possible Link to the Etiology of Eating Disorders"

_microorganisms, 2020, doi:10.3390/microorganisms8040530_

Round 1

Reviewer 1 Report

The manuscript by Breton et al. studied the effect of chronic food restriction on the production of ClpB from gut bacteria. They demonstrated that starvation and sex female determine ClpB increase. Finally, the authors suggest a link between ClpB increase in female and the development of eating disorders.The paper in interesting. However, the following comments should be addressed addressed:

-In the results (3.2 section) it is reported that ClpB increases after food restriction in both males and female rats and that plasma levels of ClpB correlated positively with fecal ClpB DNA content only in males. How the authors can explain this result?

-It is not clear why plasma levels of ClpB are not correlated with fecal ClpB DNA in females (Figure 2E).

-Which are the plasma levels of ClpB in female ABA mice?

- The 3.3 section and the relative Figure are difficult to read. It should be reorganized in a clearer manner.

Author Response

Reviewer 1

The manuscript by Breton et al. studied the effect of chronic food restriction on the production of ClpB from gut bacteria. They demonstrated that starvation and female sex determine ClpB increase. Finally, the authors suggest a link between ClpB increase in female and the development of eating disorders. The paper is interesting. However, the following comments should be addressed:

Response: We thankful to the Reviewer for the appreciation of our study and the comments which are addressed below.

-In the results (3.2 section) it is reported that ClpB increases after food restriction in both males and female rats and that plasma levels of ClpB correlated positively with fecal ClpB DNA content only in males. How the authors can explain this result?

Response: Our understanding of this phenomenon is that the plasma levels of bacterial ClpB are dependent on several factors including: 1) amount of ClpB-producing bacteria mainly in the gut as reflected by the fecal ClpB DNA; 2) ClpB transport across the gut barrier and 3) immune response to ClpB, which leads to its elimination from the blood. We can speculate that a linear relation between plasma levels of ClpB and ClpB DNA found in male but not female rats can be related to a more variable immune response in females vs. males, ex. females characterized by increased basal levels of total immunoglobulins (Tennoune et al 2015) which production is dependent on the estrous cycle (Nardelli-Haefliger et al. 1999). We cannot exclude either existence of sex differences at the level of the ClpB transport across the intestinal barrier, which may affect the linearity. This response was included in the discussion.

References:

Tennoune, N., Legrand, R., Ouelaa, W., Breton, J., Lucas, N., Bole-Feysot, C., Rego, J.-C. D., Déchelotte, P. & Fetissov, S. O. 2015. Sex-related effects of nutritional supplementation of Escherichia coli: Relevance to eating disorders. Nutrition, 31, 498-507.

Nardelli-Haefliger D, Roden R, Balmelli C, Potts A, Schiller J, De Grandi P. Mucosal but not parenteral immunization with purified human papillomavirus type 16 virus-like particles induces neutralizing titers of antibodies throughout the estrous cycle of mice. Journal of virology. 1999;73(11):9609-13.

-It is not clear why plasma levels of ClpB are not correlated with fecal ClpB DNA in females (Figure 2E).

Response: see above

-Which are the plasma levels of ClpB in female ABA mice?

Response: In the ABA experiment we studied only male mice.

- The 3.3 section and the relative Figure are difficult to read. It should be reorganized in a clearer manner.

Response: The Figure 3 has been reorganized for clarity and the corresponding changes in the Figure legend were added.

Reviewer 2 Report

The manuscript  deals  with important and interesting  topics. The authors tried to contribute to the etiopathogenetic mechanisms of eating disorders. In their previous published studies  they have shown that enterobacterial caseinolytic protease B (ClpB) could be an antigen mimetic of anorexigenic neuropeptide (alfa-melanocyte-stimulating hormone). In present study they tried to demonstrate  that starvation and female sex of rats increases enterobacterial ClpB production.

The manuscript is well written and understandable however it has some deficiencies which have to be improved.

Generally, for real proof of the changes in ClpB production under starvation and sex effects the analyses of whole gut microbiota composition will be needed. Detection of increasing abundance of enterobacteriacea under starvation will indicate that this effect is mediated by changes in gut microbiota composition.

In Materials and Methods the Biotyper bacterial protein chromatography technology is not described.  

It is necessary to explain why the authors used bacterial cultures for studying direct  effect of estradiol or testosteron on ClpB production in vitro. Is it known that these hormones are naturally present in gut lumen to enable interaction with bacteria like in the experiment described in this manuscript?

Results: In Figure 2B the deviations are missing.

Discussion: I would recommend  to add some explanation about the possible mechanisms how "increased levels of anti ClpB antibodies may at least partly compensate for increased ClpB production and hence reduce ClpB-induced satiety signaling". Moreover I would recommend to add few sentences with references about the involvement of commensal bacteria in etiopathogenesis of various autoimmune diseases and the possible mechanisms involved e.g. :

Tlaskalová-Hogenová H, Stepánková R, Hudcovic T, Tucková L, Cukrowska B, Lodinová-Zádníková R, Kozáková H, Rossmann P, Bártová J, Sokol D, Funda DP, Borovská D, Reháková Z, Sinkora J, Hofman J, Drastich P, Kokesová A.Commensal bacteria (normal microflora), mucosal immunity and chronic inflammatory and autoimmune diseases. Immunol Lett. 2004 May 15;93(2-3):97-108

Round JL, Mazmanian SK.The gut microbiota shapes intestinal immune responses during health and disease. Nat Rev Immunol. 2009 May;9(5):313-23.

Sharon G, Sampson TR,  Geschwind DH, Mazmanian SK. The Central nervous System and the Gut Microbiome. Cell. 2016 Nov 3;167(4):915-932.

Author Response

The manuscript deals with important and interesting topics. The authors tried to contribute to the etiopathogenic mechanisms of eating disorders. In their previous published studies they have shown that enterobacterial caseinolytic protease B (ClpB) could be an antigen mimetic of anorexigenic neuropeptide (alfa-melanocyte-stimulating hormone). In present study they tried to demonstrate that starvation and female sex of rats increases enterobacterial ClpB production.

The manuscript is well written and understandable however it has some deficiencies which have to be improved.

Response: We are grateful to the Reviewer for the appreciation of our study and the comments which are addressed below.

- Generally, for real proof of the changes in ClpB production under starvation and sex effects the analyses of whole gut microbiota composition will be needed. Detection of increasing abundance of Enterobacteriaceae under starvation will indicate that this effect is mediated by changes in gut microbiota composition.

Response: analyzis of gut microbiota composition could be in fact an alternative way to study the effect of starvation on relative abundance of ClpB-producing bacteria, which in the case of an increase can be interpreted as an increased ClpB production. Our view, however, is that an absolute level of ClpB DNA is the best way to characterize its production which is independent on gut microbiota composition. In fact, an absolute increase of ClpB DNA can be present even in the case of a lower relative abundance of Enterobacteriaceae when bacterial diversity is decreased.

- In Materials and Methods the Biotyper bacterial protein chromatography technology is not described. 

Response: The following information has been added to the M&M section

Identification of bacteria by MALDI-TOF MS Biotyper

Bacterial strains from fecal microbiota of male and female rats, before and after restriction, were isolated on Luria-Bertani medium and identified by analysis of the total proteome using an Autoflex III Matrix-Assisted Laser Desorption/ Ionization-Time-Of-Flight mass spectrometer (MALDI-TOF MS) (Bruker, Marcy-l’Etoile, France) coupled to the MALDI-Biotyper 3.1 system, as previously described (Hillion et al., 2013; Zommiti et al., 2018). Formic acid was used on the bacterial spots as a quick extraction procedure (Haigh et al., 2011), then the MALDI target plate was introduced in the mass spectrometer for measurement and data acquisition. For each sample, 600 spectra were pooled, and the generated spectra were compared with the MALDI-Biotyper 3.1 database. A score was calculated based on the matching between the reference spectrum and the unknown spectrum. A score of ≥ 2.0 allows species identification (Sogawa et al., 2011).

References

Haigh, J., Degun, A., Eydmann, M., Millar, M., and Wilks, M. (2011). Improved performance of bacterium and yeast identification by a commercial matrix-assisted laser desorption ionization–time of flight mass spectrometry system in the clinical microbiology laboratory. J. Clin. Microbiol. 49, 3441. doi: 10.1128/JCM.00576-11

Hillion, M., Mijouin, L., Jaouen, T., Barreau, M., Meunier, P., Lefeuvre, L., et al. (2013). Comparative study of normal and sensitive skin aerobic bacterial populations. Microbiologyopen 2, 953–961. doi: 10.1002/mbo3.138

Sogawa, K., Watanabe, M., Sato, K., Segawa, S., Ishii, C., and Miyabe, A. (2011). Use of the MALDI BioTyper system with MALDI-TOF mass spectrometry for rapid identification of microorganisms. Anal. Bioanal. Chem. 400, 1905–1911. doi: 10.1007/s00216-011-4877-7

Zommiti, M., Cambronel, M., Maillot, O., Barreau, M., Sebei, K., Feuilloley, M.G., et al. (2018) Evaluation of probiotic properties and safety of Enterococcus faecium isolated from artisanal Tunisian meat ‘Dried Ossban’. Front. Microbiol. 9:1685. doi: 10.3389/fmicb.2018.01685

It is necessary to explain why the authors used bacterial cultures for studying direct effect of estradiol or testosterone on ClpB production in vitro. Is it known that these hormones are naturally present in gut lumen to enable interaction with bacteria like in the experiment described in this manuscript?

Response: By in vitro study of the effects of sex hormones on ClpB production we indeed intended to model the in vivo situation. In fact, the concentrations of sex hormones were selected based on their physiological levels in humans. Sex hormones are naturally present in the gut and play a physiological role in regulation of microbiome action on the host (Baker at al 2017)

Reference

BAKER, J. M., AL-NAKKASH, L. & HERBST-KRALOVETZ, M. M. 2017. Estrogen-gut microbiome axis: Physiological and clinical implications. Maturitas, 103, 45-53.

Results: In Figure 2B the deviations are missing.

Response: The Figure 2 B actually shows the number of animal in which at least 1 colony has been found, so that it teaches 9 or 10 the maximal number of the animals in the group.

Discussion: I would recommend to add some explanation about the possible mechanisms how "increased levels of anti ClpB antibodies may at least partly compensate for increased ClpB production and hence reduce ClpB-induced satiety signaling". Moreover I would recommend to add few sentences with references about the involvement of commensal bacteria in etiopathogenesis of various autoimmune diseases and the possible mechanisms involved e.g. :

Tlaskalová-Hogenová H, Stepánková R, Hudcovic T, Tucková L, Cukrowska B, Lodinová-Zádníková R, Kozáková H, Rossmann P, Bártová J, Sokol D, Funda DP, Borovská D, Reháková Z, Sinkora J, Hofman J, Drastich P, Kokesová A. Commensal bacteria (normal microflora), mucosal immunity and chronic inflammatory and autoimmune diseases. Immunol Lett. 2004 May 15;93(2-3):97-108

Round JL, Mazmanian SK.The gut microbiota shapes intestinal immune responses during health and disease. Nat Rev Immunol. 2009 May;9(5):313-23.

Sharon G, Sampson TR,  Geschwind DH, Mazmanian SK. The Central nervous System and the Gut Microbiome. Cell. 2016 Nov 3;167(4):915-932.

Response: Thank you for this suggestion, the relevant discussion has been extended.

Round 2

Reviewer 1 Report

The manuscript by Breton et al. in the revised form can be accepted for publication.

Author Response

We are grateful to the Reviewer for the appreciation of our study and the comments which are addressed below.